# Dual Insecticidal Effects of *Adenanthera pavonina* Kunitz-Type Inhibitor on *Plodia interpunctella* is Mediated by Digestive Enzymes Inhibition and Chitin-Binding Properties

**DOI:** 10.3390/molecules24234344

**Published:** 2019-11-28

**Authors:** Caio Fernando Ramalho de Oliveira, Taylla Michelle de Oliveira Flores, Marlon Henrique Cardoso, Karen Garcia Nogueira Oshiro, Raphael Russi, Anderson Felipe Jácome de França, Elizeu Antunes dos Santos, Octávio Luiz Franco, Adeliana Silva de Oliveira, Ludovico Migliolo

**Affiliations:** 1Universidade Federal de Grande Dourados, Dourados, Mato Grosso do Sul, MS, 79825-070, Brazil; oliveiracfr@gmail.com; 2S-Inova Biotech, Programa de Pós-Graduação em Biotecnologia, Universidade Católica Dom Bosco, Campo Grande, MS, 79117-900, Brazil; taylla.flores@outlook.com (T.M.d.O.F.); marlonhenrique6@gmail.com (M.H.C.); oshiro.kgn@gmail.com (K.G.N.O.); ocfranco@gmail.com (O.L.F.); 3Programa de Pós-Graduação em Biologia Celular e Molecular, Universidade Federal da Paraíba, João Pessoa, PB, 58059-900, Brazil; 4Centro de Análises Bioquímica e Proteômicas, Programa de Pós Graduação em Ciências Genômicas e Biotecnologia, Universidade Católica de Brasília, Brasília, DF, 70790-160, Brazil; 5Programa de Pós-Graduação em Patologia Molecular, Faculdade de Medicina, Universidade de Brasília, Brasília, DF, 70910-900, Brazil; 6Programa de Pós-Graduação em Bioquímica, Universidade Federal do Rio Grande do Norte, Natal, RN, 59078-900, Brazil; raphaelrussi@gmail.com (R.R.); andersonfjf@gmail.com (A.F.J.d.F.); elizeu.ufrn@gmail.com (E.A.d.S.); cisteana@yahoo.com.br (A.S.d.O.)

**Keywords:** non-competitive inhibitor, trypsin inhibitor, peritrophic membrane

## Abstract

The Indianmeal moth, *Plodia interpunctella*, is one of the most damaging pests of stored products. We investigated the insecticidal properties of ApKTI, a Kunitz trypsin inhibitor from *Adenanthera pavonina* seeds, against *P. interpunctella* larvae through bioassays with artificial diet. ApKTI-fed larvae showed reduction of up to 88% on larval weight and 75% in survival. Trypsin enzymes extracted from *P. interpunctella* larvae were inhibited by ApKTI, which also demonstrated capacity to bind to chitin. Kinetic studies revealed a non-competitive inhibition mechanism of ApKTI for trypsin, which were further corroborated by molecular docking studies. Furthermore, we have demonstrated that ApKTI exhibits a hydrophobic pocket near the reactive site loop probably involved in chitin interactions. Taken together, these data suggested that the insecticidal activity of ApKTI for *P. interpunctella* larvae involves a dual and promiscuous mechanisms biding to two completely different targets. Both processes might impair the *P. interpunctella* larval digestive process, leading to larvae death before reaching the pupal stage. Further studies are encouraged using ApKTI as a biotechnological tool to control insect pests in field conditions.

## Highlights

ApKTI increases mortality of *P. interpunctella* larvae.In vitro analyses showed that ApKTI presents chitin-binding properties.In silico structural studies corroborated protein-protein and protein–carbohydrate interactions.

## 1. Introduction

Insect pests are responsible for immense losses in the field and storage conditions. Annually, billions of dollars are spent in crop protection, mainly with highly toxic insecticides [1]. As consequence of the massive use of chemical insecticides, the reduction of natural biodiversity near crops and the selection of resistant insect pest have been reported [2]. The overwhelming resistance of insects against chemical pesticides encourages the investments on alternative control strategies, including pyramiding of genes with insecticidal activity [1,3,4]. *Plodia interpunctella* (Hübner) (Lepidoptera: Pyralidae) is a widely distributed insect pest from temperate and tropical areas [3]. During its larval stage, *P. interpunctella* prompts significant economic impacts on stored foods, such as cereals, legumes, dried fruits, and nuts [5].

Studies regarding the digestion process in insects, the gut morphology, compartmentalization, and function have emerged features which might be applied in pest control strategies [6,7]. Therefore, the use of molecules to impair the functional digestive physiology might impact insects’ development with direct consequences on both weight and survival. For instance, most Lepidopteran present serine-peptidases, especially trypsin and chymotrypsin, as major enzymes for initial digestion of dietary proteins. Thus, the use of molecules capable of interfering with proteolysis raises diverse impacts on larval development.

Diverse plant proteins display insecticide activity. The plant defense mechanisms against insects are result of a co-evolution of hundreds of millions of years [8]. Among the proteinaceous compounds, plant peptidase inhibitors (PIs) appear as a promising group for heterologous expression in crops. PIs can also be found in different plant tissues, contributing to plant defense against herbivory through the inhibition of insect gut peptidases, reducing the availability of amino acids necessary for growth and development [9]. Genes encoding plant PIs for the transformation of crops have been reported as an alternative to control insect pests [10,11,12].

Some PIs bind to chitin [5,13], a polysaccharide present in multiple structures and also at insects’ peritrophic membrane (PM). The PM is an anatomical structure that surrounds the food bolus in the insect gut, displaying a dual function: (i) Compartmentalization of the digestive process; and (ii) facilitation of the food bolus movement [7]. The binding or interference in PM homeostasis impairs nutrient absorption, decreasing the larval weight and increasing the mortality [5,13]. Therefore, the PM has been considered an important target for insecticidal agents’ development, and the consumption of PIs by insect larvae can be used to affect the PM homeostasis.

The Kunitz trypsin inhibitor, ApKTI, has been isolated from *Adenanthera pavonina* Linnaeus seeds (Fabaceae: Mimosoideae) [14]. ApKTI consists of a double polypeptide chain and is capable of inhibiting two different classes of peptidases, including serine- and cysteine-peptidases [15]. Thus, studies have shown that ApKTI is effective in controlling insect pests from different orders, including Lepidoptera [16,17], Coleoptera [9], and Diptera [18]. Based on that, here we aimed to investigate the effects of ApKTI on *P. interpunctella* larval development. The insecticide properties of ApKTI were investigated in vivo, followed by in vitro assays and bioinformatics studies to determine ApKTI’s possible targets in *P. interpunctella* larval gut. Finally, we used molecular docking to investigate the binding mode of ApKTI on trypsin, chymotrypsin, and N-acetylglucosamine (GlcNAc—the elementary unit of chitin), explaining how ApKTI binds to chitin and inhibits digestive enzymes, influencing negatively the *P. interpunctella* development.

## 2. Results

### 2.1. Bioassays with P. Interpunctella Larvae

We carried out bioassays to investigate the potential of ApKTI as insecticide agent. Artificial diets were prepared with different ApKTI concentrations (0.3%–1.5% *w*/*w*) and offered to *P. interpunctella* neonate larvae. Following a chronic exposure to ApKTI over 15 days, the larval weight, survival and enzymatic activity were analyzed. ApKTI-fed larvae presented a clear dose-dependent reduction on both weight and survival. The highest concentration of ApKTI into artificial diet resulted in a reduction of 88% and 75% on larval weight and survival, respectively (Figure 1A,B). Through bioassays we demonstrated that ApKTI showed insecticide activity against *P. interpunctella* larvae (Figure 1C).

### 2.2. Purification of Trypsin Inhibitor from Adenanthera Pavonina Seeds (ApKTI)

Initially, a precipitation with ammonium sulfate was used to fractionate the crude extract into fractions. Afterwards, the inhibitory activity against trypsin was determined to each fraction. We observed that the fraction 40%–60% from ammonium sulfate precipitation showed the highest inhibitory activity against trypsin. This fraction was chosen to further purification of ApKTI. First, the fraction 40%–60% was applied into gel filtration chromatography, resulting in two peaks: (i) A first peak corresponding to high molecular mass proteins and; (ii) a second peak containing a mix of intermediate and low molecular mass proteins. The fractions collected at the final portion of the second peak revealed trypsin inhibitory activity (Figure 2A). These active fractions were pooled and applied onto an affinity trypsin-Sepharose column (Figure 2B). The chromatogram of trypsin-Sepharose column showed proteins eliminated during the washing step with no affinity by column. The elution of fractions adsorbed to column, started from fraction 20, resulted in elution of a single peak. This peak showed inhibitory activity against trypsin and, therefore, was named ApKTI (Figure 2B).

A chromatography using chitin as matrix was used to investigate whether ApKTI possesses chitin-binding properties. Chitin was used since this polysaccharide is a major constituent of PM, being a possible target of ApKTI into larval gut. ApKTI was injected into the chitin column and no peaks were detected during the column-washing step. Following the elution step using HCl, a single peak was noticed (Figure 3). Further assays confirmed that the eluted peak showed inhibitory activity against trypsin (data not shown). Based in this result, we demonstrated that ApKTI possesses chitin-binding properties.

### 2.3. In Vitro Enzymatic Assays

Following the studies regard chitin-binding properties, enzymatic assays using *P. interpunctella* gut extract as source of enzymes were carried out to investigate the effects of ApKTI on inhibition of digestive enzymes, another possible target into larval gut. For detection of inhibitory activity against trypsin we used the specific substrate BApNA. The gradual increasing of ApKTI concentration led to reduction of trypsin activity of *P. interpunctella* gut extract. The maximal inhibitory activity (90.6%) was obtained in the presence of 15 μg ApKTI. Moreover, a kinetic study carried out with different concentrations of ApKTI and BApNA revealed a non-competitive inhibition mechanism of ApKTI for *P. interpunctella* trypsin (Figure 4). Dixon Plot revealed that lines corresponding to the concentrations of the substrates converge to a common point in the *X*-axis. The *Ki* value was obtained (504.7 nM) after determination of *Km* and *V_max_*, in agreement with the *Ki* of other non-competitive PIs [19]. These results indicate a clear trypsin inhibitory activity based in a non-competitive mechanism. To gain insights this result a molecular level, were further investigated the binding of ApKTI with chitin and serine-peptidases through in silico experiments.

### 2.4. Structural Studies

After modeling, validation procedures and tridimensional fold checking, we investigated the molecular mechanism of ApKTI binding to GlcNAc, a monosaccharide that constitutes the chitin polymer. The affinity value obtained for ApKTI/[(GlcNAc)_3_] was −8.9 kcal.mol^−1^. As summarized in Appendix A, all predicted atomic interactions were characterized as hydrogen bonds (HB), with a small number of interactions with larger distances were predicted (6 HBs).

For complex with GlcNAc, Arg^64^ as well P2 residue from the reactive site loop of ApKTI, Arg^66^, were involved in stabilization, beyond the amino acid residues Glu^77^, Thr^75^, and Gln^112^ (Figure 5; Appendix A). These residues are near a hydrophobic patch, where part of the GlcNAc structure anchored. Regarding the ApKTI binding ability to GlcNAc, the involvement of the reactive site loop region flanked by a hydrophobic patch connecting antiparallel β-strand arrangement in ApKTI is involved in the insertion of GlcNAc. Therefore, through in silico results we suggested the mechanism and molecular region of ApKTI involved with binding to chitin.

### 2.5. Structural Studies

We carried out further molecular modeling and docking simulations to obtain insights on the binding mode of the molecular complexes ApKTI/trypsin and ApKTI/chymotrypsin. Molecular modeling simulations were carried out to predict the atomic coordinates of ApKTI, trypsin and chymotrypsin, thus generating reliable tridimensional structures for docking studies. The lowest free-energy models were selected and validated as described in the methodology. All models generated by comparative modeling presented >80% amino acid residues in the most favorable regions in the Ramachandran Plot. Moreover, the calculated overall G-factors (average scores for the dihedral angles and the main-chain covalent forces) ranged from −0.23 to −0.26 for all models here studied, indicating reliable structures regarding stereochemical parameters (reference value: >0.5). The fold quality of each model was also confirmed, with calculated z-scores of −4.8, −6.52, and −6.47 for ApKTI, trypsin and chymotrypsin, respectively (Table 1). Structurally, ApKTI theoretical model adopted a loop connecting β-strand scaffold with two polypeptide chains stabilized by two disulfide bonds. The ApKTI reactive site loop is composed by Thr, Pro, Arg, Ile, Tyr, and Gly amino acid residues (Figure 6A,C). The theoretical models for trypsin and chymotrypsin from *P. interpunctella* showed two main cores formed by antiparallel β-strands, as well as α-helix and loop arrangements (Figure 6A,C).

After modeling, validation procedures and tridimensional fold checking, ApKTI, trypsin, and chymotrypsin were submitted to molecular docking studies. In this context, two molecular complexes were here analyzed. The best affinity values obtained for ApKTI/trypsin, and ApKTI/chymotrypsin were −13.3 and −13.4 kcal.mol^−1^, respectively. As summarized in Appendix A, all predicted atomic interactions were characterized as hydrogen bonds (HB), with distances between 2.7 to 3.6 Å. A total of 7 HB were predicted for ApKTI/trypsin; whereas 10 HB could be calculated for the ApKTI/chymotrypsin complex. However, even though ApKTI/chymotrypsin presents a higher number of interactions, as well as a slightly better affinity, the distances of interactions for the complex ApKTI/trypsin appear to be shorter, which might explain its similar binding affinity with ApKTI/chymotrypsin. In complexes with enzymes, crucial amino acid residues such as Arg^64^, located at position P1 from the reactive site loop of ApKTI, and Arg^138^, were always involved in the stabilization of the complexes (Figure 6B,D; Appendix A).

Furthermore, we also described a non-competitive inhibition mechanism between ApKTI and serine-peptidases, corroborating in vitro assays, once that no interactions were predicted between ApKTI reactive site loop and the active site of trypsin and chymotrypsin. The binding between ApKTI and enzymes makes it difficult for substrates to occupy the enzymatic active site, reducing enzymatic catalysis in the presence of the inhibitor. Taken together, in silico results supported the non-competitive inhibition mode of ApKTI for trypsin and chymotrypsin.

## 3. Discussion

Currently, most efforts for crop protection reside in both *Bt* technology and massive use of insecticides. However, the events of insect resistance against insecticides and *Bt* crops are overwhelming reported [2,23,24]. The consequence of systematic insecticide application resides on insect resistance and lethal effects on non-target organisms, including parasites, parasitoids, and pollinators. Moreover, insecticides also represent a huge threat to the environment and human health. For this reason, alternative tools to reduce losses in field have been investigated. The expression of plant defense proteins in crops has been studied since the late 1980s [11,12,25,26]. Plant defense proteins, including ApKTI, are result of insect-plant coevolution of millions of years [27]. Naturally, PIs are expressed in diverse plant tissues, including tubers, leaves, flowers, and seeds [28]. Several crops consumed by humans are rich in PIs, such as soybean, beans, tomato, potato, and legume seeds in general [29], reaffirming that the introduction of PIs coding genes in plants is more likely not to compromise human health. However, prior the development of transgenic plants expressing PIs, extensive studies have to be carried out to evaluate the effects of a PI on target insects, as insect pests are known to overcome the insecticide effects of PIs by modulating digestive enzymes expression [30,31,32,33].

The purification of ApKTI was firstly reported in 1986 by Richardson and collaborators [14]. Since then, the insecticidal activity of ApKTI has been investigated against Coleoptera [9] and Lepidoptera [16,17]. The ability of ApKTI to inhibit enzymes from different classes has been previously studied [15]. However, to the best of our knowledge is the first time that the binding of ApKTI to chitin has been suggested. Although it seems new, the description of multi-activity plant proteins is already described in the literature. Several examples include storage proteins with chitin-binding properties [34], plant inhibitors with antimicrobial [35] and antiparasitic activities [36]. These finding demonstrated that plant proteins, including PIs, might assume a similar fold with a diversity of biological functions. *Talisia esculenta* (Sapindaceae) seeds contain an insecticide storage protein named Talisin, that shows both lectin-like properties and inhibitory activities [37]. A Kunitz inhibitor purified from *Entada acaciifolia* seeds, named EATI, presented inhibitory activity against trypsin [38]. Despite its high inhibitory affinity for trypsin, we recently observed through immunofluorescence assays that EATI also binds to peritrophic membrane of a lepidopteran larvae, *Spodoptera frugiperda* (data not published). The literature point that the defensive functions of PIs are not restricted to insect enzyme inhibition [39].

The importance of ApKTI binding to chitin is related to PM composition, given the significance of PM in protection against mechanical damage, microorganisms and parasites invasion [7]. The PM integrity is directly involved in high digestive efficiency observed among insects, as it creates compartments for initial, intermediate and final digestion and serves as site to anchorage digestive enzymes [7]. Our results show that ApKTI acts through enzymatic inhibition and chitin binding, being able to trigger PM perturbations.

Is important to note that in our bioassays with *P. interpunctella* larvae, the concentrations of ApKTI were similar to reports carried out with PIs incorporated in artificial diets [38,40] or even transgenic plants expressing PIs [11,12]. In a similar study, Amorim and collaborators [5] demonstrated that SBTI, a competitive trypsin and chymotrypsin inhibitor from soybean, was ineffective against *P. interpunctella*. SBTI-fed larvae presented neither mortality nor weight reduction when an artificial diet was supplemented up to 4% of SBTI. The differences between ApKTI and SBBI effects on *P. interpunctella* might be related to their different enzymatic inhibition mechanisms. SBBI is a competitive inhibitor, therefore, its reactive site forms a tight complex with the enzyme active site, which composed by Asn, His and Ser residues. ApKTI is a non-competitive inhibitor, thus, its reactive site binds to residues located out of the enzyme active site. For trypsin, ApKTI binds to Gly^50^ and neighbor amino acids; for chymotrypsin, ApKTI binds to a region near Leu^238^ (Appendix A). The adaptation of Lepidopteran to serine-peptidases is a phenomenon well described [31,32,41]. In general, a differential transcription of enzymes with a reduced affinity for plant PIs allows the insects to perform protein digestion without interference of ingested PIs. The resistant enzymes present amino acid substitutions of conserved residues involved in interactions between the enzyme active site and PIs reactive loop [42,43]. It results in a reduced affinity between resistant enzymes and competitive PIs. The evolution selected subsites in insects’ trypsin progressively more hydrophobic, whereas plant PIs present polar residues at their reactive site [44,45]. Considering that ApKTI is a non-competitive inhibitor, the affinity for trypsin is not expected to be affected by amino acid substitution at the enzyme active site. This feature represents benefits over competitive inhibitors.

The modeling of ApKTI showed twelve antiparallel β-sheets along two polypeptide chains, a structural fold previously reported [15] and well described for other Kunitz-type inhibitors [42]. The theoretical models for trypsin and chymotrypsin from *P. interpunctella* are also in agreement with serine-peptidases characterized previously [18]. These findings have been described in previous reports carried out with Kunitz-type inhibitors [18,46,47,48], supporting the accuracy of our in silico data. Moreover, the distances of hydrogen bonds between ApKTI/trypsin (2.7–3.6 Å) and ApKTI/chymotrypsin (3.0–3.6 Å) are in agreement with data from the literature for docking studies with serine peptidases and PIs [15,49]. Moreover, we demonstrated that hydrogen bonds play a crucial role in ApKTI stabilization with trypsin, chymotrypsin and GlcNAc.

Our bioassays showed that ApKTI displays insecticide activity on *P. interpunctella* larvae. Moreover, we propose that ApKTI might affects the *P. interpunctella* larvae development by two distinct mechanisms: (i) Inhibition of digestive enzymes and (ii) binding to chitin present in PM. The inhibition of a fraction of digestive enzymes impair the digestive process, with direct effects on larval performance. The binding of ApKTI to chitin might impairs the functioning of PM, such as obstructing the endo-ectoperitrophic circulation of food and enzymes, reducing the digestive efficiency [47] or the enzyme recycling. However, a conclusive finding would be obtained through ultrastructural study targeting the PM integrity.

## 4. Material and Methods

### 4.1. Chemicals

N-benzoyl-D-L-arginine-p-nitroanilide (BApNA), bovine serum albumin (BSA), bovine pancreatic trypsin, Trypsin-chymotrypsin inhibitor from *Glycine max* (SBBI) and Chitin from shrimp shells were purchased from Sigma Chemical Co. (St. Louis, MO, USA). All other chemicals and reagents used were of analytical grade.

### 4.2. Insects

Larvae of *P. interpunctella* were maintained in the Department of Genetics and Cell Biology, Bioscience Center, Federal University of Rio Grande do Norte, Natal, Brazil, and fed with artificial diet (10.4% finely ground sugar cane fibers, 3% wheat germ, 6.5% wheat flour, 12% crystal sugar, 9.9% yeast, 0.3% sodium benzoate, 0.9% HCl and 57% H_2_O), at 25–30 °C and relative humidity of 70%–80% [5].

### 4.3. Bioassays with *Plodia Interpunctella* Larvae

The effect of ApKTI on *P. interpunctella* larval development was investigated in bioassays using artificial diet, based on a mixture of wheat flour (2.3 g), brewer’s yeast (2.5 g), white sugar (3.0 g), and sodium benzoate (0.07 g) for a total of 8 g of the mixture. The ingredients were homogenized for obtainment of fine flour. Artificial diets with different ApKTI concentrations were prepared, 0.3%; 0.5%; 1.0%; and 1.5% (*w*/*w*). The control group had the artificial diet enriched with BSA at the same concentrations used for ApKTI. The artificial diets (800 mg) were placed individually in in 6-well microplates and each well received nine neonatal larvae and the plates wrapped with clear PVC film were maintained at temperature of 25 ± 1.0 °C, relative humidity of 60%–70% during 15 days. After this period, the larvae were weighed for determination of average larval weight and survival. The bioassay was carried out in triplicate.

### 4.4. Purification of Trypsin Inhibitor from *Adenanthera Pavonina* Seeds (ApKTI)

*A. pavonina* seeds were obtained locally (Campo Grande-MS) and stored at −20 °C until use. The *A. pavonina* fine-grained flour from seeds was obtained and subjected to extraction with sodium tetraborate 50 mM buffer, pH 7.5, in the ratio 1:10 (flour: buffer), during 3 h under stirring at room temperature. After, it was centrifuged at 12,000× *g*, for 30 min, at 4 °C. The supernatant was submitted to a process of precipitation with ammonium sulfate. Three fractions of ammonium sulfate were prepared: 0%–40%, 40%–60% and 60%–90% of saturation. After each saturation step the solution remained at 4 °C for approximately 20 h and then was centrifuged at 12,000× *g*, for 30 min, at 4 °C. The precipitate resulted from each step were resuspended in sodium tetraborate 50 mM buffer (pH 7.5) and subjected to dialysis at the same buffer.

The 40%–60% fraction, which showed the highest inhibitory activity against trypsin, was applied to a size exclusion chromatography SephacrylS-100 high resolution (1.50 × 115 cm) equilibrated with Sodium tetraborate 50 mM buffer, pH 7.5, at flow of 1 mL.min^−1^. Fractions that showed inhibitory activity against trypsin were pooled and applied on to affinity column trypsin-Sepharose (2.5 × 2.0 cm) previously equilibrated with sodium tetraborate 50 mM buffer, pH 7.5. The proteins retained in the resin were eluted using 100 mM HCl solution at a flow of 40 mL.h^−1^. The absorbance was monitored at 280 nm. The fractions with inhibitory activity against trypsin were dialyzed in distillated water, lyophilized, and named ApKTI. All fractions had the protein content measured by Bradford [50] using bovine serum albumin (BSA) as standard.

### 4.5. Chromatography in Chitin Column

A chitin column (2.0 × 2.5 cm) was prepared and washed with HCl 10 mM, subsequently equilibrated with sodium tetraborate 50 mM buffer, pH 7.5. ApKTI was applied to column and the elution carried out with HCl 100 mM. Fractions of 2 mL were collected, and the absorbance was monitored at 280 nm.

### 4.6. Kinetic Studies Between ApKTI and *P. Interpunctella* Trypsin

Kinetic studies were carried out in order to investigate the mechanism of inhibition of ApKTI on trypsin from *P. interpunctella.* Three different concentrations of BApNA (0.31; 0.62; and 1.25 mM) and increasing concentrations of ApKTI (0.25; 0.5; 1; 3; 6; 9; and 12 µg) were incubated with a fixed concentration of *P. interpunctella* gut extract. Followed the establishment of kinetic parameters *V_max_* and *Km*, the *Ki* (dissociation constant) values were determined.

### 4.7. In Vitro Evaluation of Inhibitory Activity

#### 4.7.1. Inhibitory Activity for Bovine Trypsin

Assays to determine the inhibitory activity of trypsin were conducted using BApNA as substrate [51]. Ten microliters of trypsin solution (0.3 mg·mL^−1^ prepared in 2.5 mM HCl) was incubated for 10 min at 30 °C with 100 µL of ApKTI, 120 µL of 2.5 mM HCl and 270 µL of 50 mM Tris-HCl buffer, pH 7.5. Followed the addition of 500 µL of 1.25 mM BApNA solution, prepared in 1% (*v*/*v*) DMSO and 0.05 M Tris-HCl, pH 7.5. The reaction occurred during 15 min at 30 °C, being stopped by the addition of 150 µL of 30% acetic acid solution. The formation of p-nitroaniline was measured by absorbance at 410 nm. All assays were made in triplicate.

#### 4.7.2. Insect Gut Peptidases

A total of 50 *P. interpunctella* fifth-instar larvae were cold-immobilized on ice and had the gut surgically removed into an iso-osmotic saline (150 mM NaCl) solution. After homogenization, the gut tissue was centrifuged at 10,000× *g*, at 4 °C, for 10 min. The supernatants were then recovered, had the protein content determined and were used as source of peptidases.

The activity of trypsin was determined using BApNA. Briefly, 20 µg of gut extract were incubated for 10 min at 30 °C Tris-HCl 50 mM buffer, pH 9.5. Further, it was added 500 µL of 1.25 mM BApNA solution, prepared in 1% (*v*/*v*) DMSO and Tris-HCl 50 mM buffer. The reaction occurred during 15 min at 30 °C, being stopped by the addition of 150 µL of 30% acetic acid solution. The formation of p-nitroaniline was measured by absorbance at 410 nm. All assays were made in triplicate.

### 4.8. Molecular Modeling

Tridimensional theoretical models for ApKTI and trypsin/chymotrypsin from *P. interpunctella* were built using Modeller v. 9.12 [52]. ApKTI primary sequence was obtained from the National Center for Biotechnology Information (NCBI) under the accession numbers gi: 124152 and gi: 124153 [14] for the α and β chains, respectively. *P. interpunctella* trypsin and chymotrypsin primary sequences were obtained from NCBI under the accession numbers gi: 3153854 [53] and gi: 2353158 [54], respectively. Trypsin and chymotrypsin signal peptides and transmembrane regions were predicted by Phobius server [55] and discarded from the simulations. ApKTI was modeled based on the crystallographic structure of a soybean trypsin inhibitor (PDB code: 1avw) [47], whereas *P. interpunctella* trypsin/chymotrypsin were modeled based on the crystallographic structure of a fire ant (*Solenopsis invicta*) chymotrypsin (PDB code: 1eq9) [56]. One hundred tridimensional theoretical models were generated and ranked according to their DOPE score. The lowest free-energy models were selected and validated by PROCHECK (stereochemical quality) [21] and ProSA-web (fold quality) [20] servers.

### 4.9. Molecular Docking

Molecular docking studies were carried out to predict the atomic interactions involved in the stabilization of the ApKTI/trypsin, ApKTI/chymotrypsin and ApKTI/N-acetylglucosamine (NAG) complexes. NAG coordinates were extracted from a hevein domain with binding affinity for chitooligosaccharides (PDB code: 1t0w) [57]. Grid boxes of 88 × 88 × 88 (ApKTI/trypsin and ApKTI/chymotrypsin) and 44 × 45 × 51 (ApKTI/NAG) points with 1 Å spacing were built on AutoDock Tools [58]. Fifty runs of molecular docking simulations were performed using AutoDock Vina [58], and the complexes ranked according to their affinities in kcal.mol^−1^. The highest affinity complexes were submitted to energy minimization steps using the GROMOS 96 46b1 force field from SPDB Viewer v. 4.1.1 [59]. Further, all atomic interactions respecting the maximum distance of 3.6 Å between all atoms (blind docking) were measured using PyMOL (http://pymol.sourceforce.net/).

## 5. Conclusions

Herein, we showed the insecticide effects of ApKTI on *P. interpunctella* larvae, demonstrating the potential this inhibitor for pest control. Based on a non-competitive inhibition mode, ApKTI was capable of compromising the larval digestive process, inhibiting trypsin and chymotrypsin enzymes. The ability of ApKTI to bind to chitin might affect the PM homeostasis, committing a variety of processes, including enzyme anchorage, compartmentalization of the gut lumen, and enzyme recycling. Together, these factors promote a limited absorption of nutrients that led ApKTI-fed larvae to death by starvation.

## Figures and Tables

**Figure 1 molecules-24-04344-f001:**
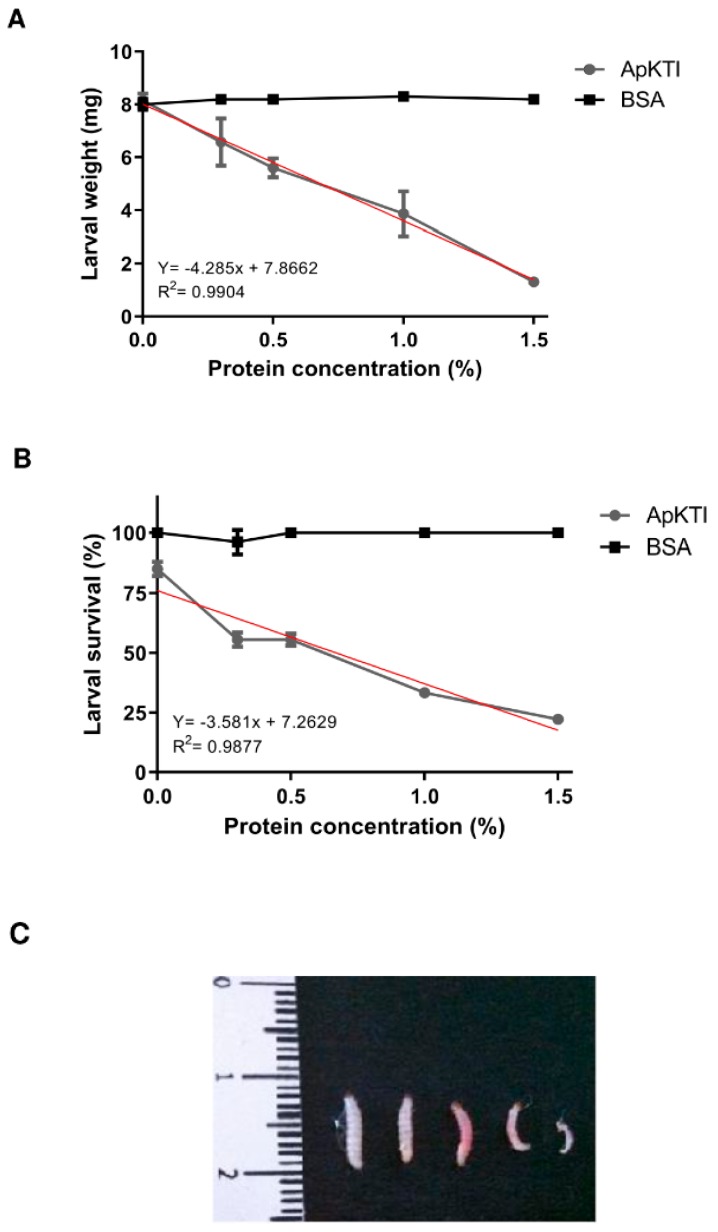
Bioassays with *P. interpunctella* larvae. Artificial diet was enriched with ApKTI from 0.3% to 1.5% (*w*/*w*). At the final of bioassays (15 days) the weight (**A**) and survival (**B**) in each treatment were determined. In control treatment the artificial diet was supplemented with bovine serum albumin (BSA) at the same concentration of ApKTI. A linear regression was obtained from ApKTI-fed larvae demonstrating the dose-response effect. (**C**) Representative image of larvae of each treatment, in sequential order from the left: control-fed larvae, larvae fed with 0.3%, 0.5%, 1.0%, and 1.5% ApKTI.

**Figure 2 molecules-24-04344-f002:**
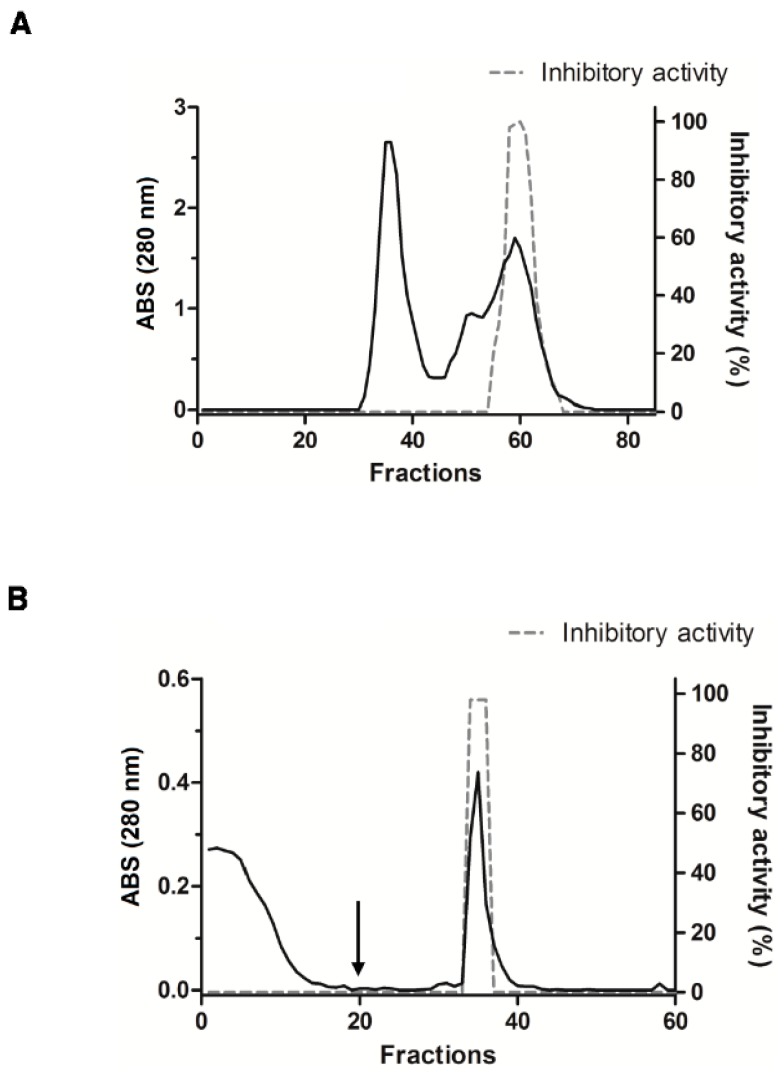
Purification of ApKTI. (**A**) The fraction F40-60% from ammonium sulfate precipitation was fractioned in Sephacryl S-100 column. (**B**) The active fractions were purified in trypsin-Sepharose column. The arrows indicate the beginning of elution with HCl 100 mM in trypsin-Sepharose and chitin columns. All fractions had the inhibitory activity against trypsin assayed. The inhibitory activities are showed as dotted lines.

**Figure 3 molecules-24-04344-f003:**
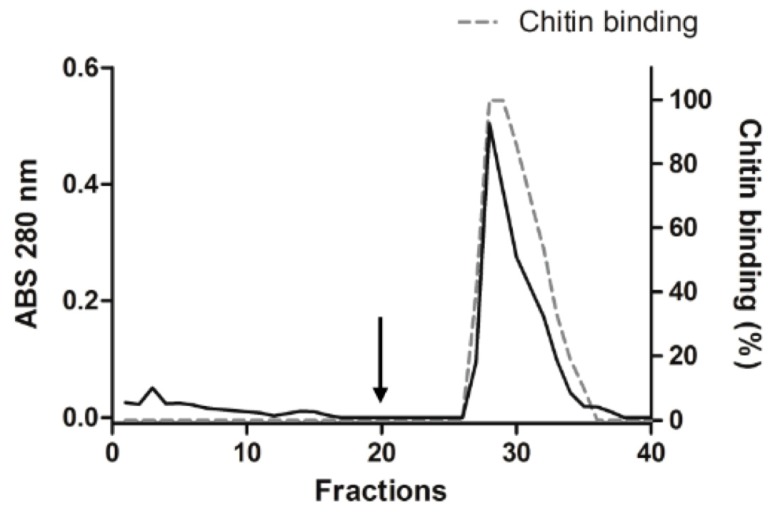
Binding of ApKTI in chitin column chromatography was used to demonstrate the affinity. The dotted line represents the binding of ApKTI in chitin column.

**Figure 4 molecules-24-04344-f004:**
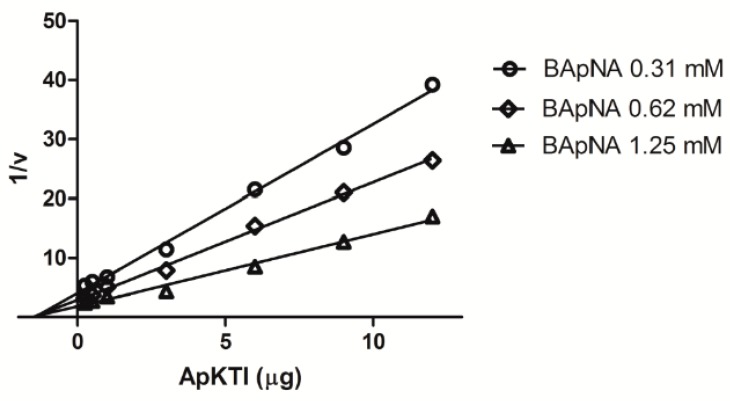
Binding mode between ApKTI and trypsin. Kinetic studies were carried out using different concentration of inhibitor and substrate. Following the Michaelis–Menten parameters determination, the Dixon plot was obtained. The points where lines intersect the *X*-axis converge to a common point, while different 1/V were observed, determining a non-competitive inhibition mechanism between ApKTI and trypsin.

**Figure 5 molecules-24-04344-f005:**
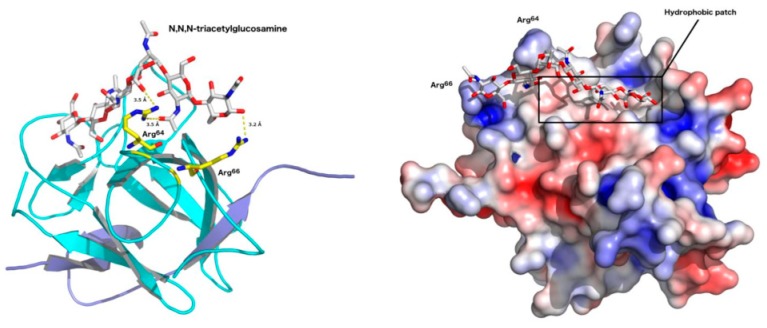
Tri-N-acetylglucosamine in complex with ApKTI, where Arg^64^ and Arg^66^ located at the reactive site loop from ApKTI are involved in the complex stabilization (left). Adaptive Poisson–Boltzmann solver (APBS) electrostatic potential of ApKTI (potential ranges from −5 kT/e (red) to + 5 kT/e (blue)), highlighting the Arg^64^ and Arg^66^ residues and the hydrophobic pocket involved in GlcNAc attachment (right).

**Figure 6 molecules-24-04344-f006:**
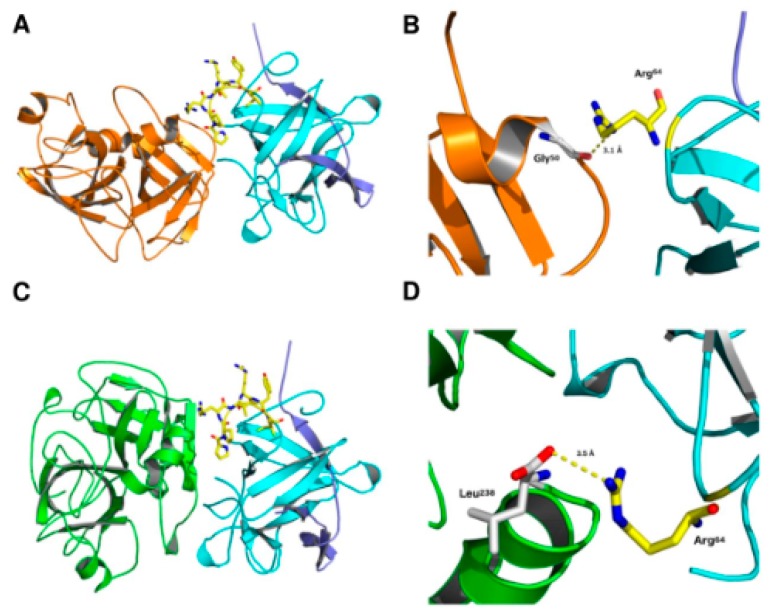
Tridimensional theoretical models for ApKTI, trypsin and chymotrypsin from *P. interpunctella*. β-strand scaffold showed two main cores formed by antiparallel β-strands, as well as α-helix and loop arrangements are observed in both enzymes. Predicted conformations for the (**A**) ApTI/trypsin and (**C**) ApTI/chymotrypsin complexes (yellow sticks represent the reactive site loop of ApTI). Atomic interactions involving Arg^64^ located at the reactive site loop from ApKTI are highlighted in the (**B**) ApKTI/trypsin and (**D**) ApTI/chymotrypsin complexes. Trypsin and chymotrypsin active sites are highlighted as white sticks.

**Table 1 molecules-24-04344-t001:** Structural statistics for the tridimensional theoretical models generated in this study for ApKTI, trypsin and chymotrypsin.

Predicted Structures	Sequence Length	Fold Quality (z-Score)	Stereochemistry (G-Factors)	Ramachandran			Bad Bonds (%)	Bad Angles (%)
				Most Favored (%)	Allowed (%)	Outliers (%)		
ApKTI	176	−4.80	−0.26	89.00	95.9	4.07	0	1.09
Trypsin	233	−6.52	−0.23	90.48	96.1	3.90	0	1.87
Chymotrypsin	238	−6.47	−0.24	91.50	97.9	2.12	0	1.70

The z-scores obtained for all structures here reported are in agreement with those with similar size, structurally determined by X-ray crystallography and deposited in the Protein Data Bank (PDB). The G-factors indicate that the overall average for the dihedral angles, along with the main-chain covalent forces for each structure are within the expected values for reliable structures (G-factors > −0.5). The structural validations were performed on ProSa-web [20], PROCHECK [21] and MolProbity [22].

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
