# Peer review of "Dual Insecticidal Effects of Adenanthera pavonina Kunitz-Type Inhibitor on Plodia interpunctella is Mediated by Digestive Enzymes Inhibition and Chitin-Binding Properties"

_molecules, 2019, doi:10.3390/molecules24234344_

Round 1

Reviewer 1 Report

I reviewed the manuscript from Oliveira et al. dealing with the identification of a chitin-binding domain within a Kunitz-type inhibitor (ApKTI), suggesting a potential dual effect of these inhibitors on insects. The authors combine structural modelling to identify this new motif with experiments aiming at characterizing the inhibitory activity of this ApKTI and its effect on lepidopteran larvae. Globally, the English ok (at least for the non-native English reader that I am) but I do believe that the organization of the manuscript should be reconsidered to make the logic more clear and the reading smoother. Efforts are notably necessary in the results section. I suggest to start with the bioassays to show the possibility of using ApKTI as a biocidal molecule. Then, ask whether it acts by enzymatic inhibition and/or chitin-binding capacity based on literature. This includes purification and testing in vitro (and in vivo – see suggestions below) to finish by structural predictions and docking analyses to highlight potential key sites.

Some clarifications/precisions/deeper analyses are required for some aspects (see detailed review below) and I believe that some functional experiments should be considered to support the conclusions. In the present state, the main weakness of the paper is to aim for identifying causal links where they so far found insights and potential correlations. With additional experiments and clarifications, I do believe that this article should be reconsidered for publication.

Molecular modeling and docking simulations should be considered with caution. All the structures used for the interaction analyses are predicted and cannot reached the level of precision of experimentally-solved ones. Authors followed a conventional approach to select the best model. However, the analyses that they are doing afterward requires a high resolution in the structures used to properly predict interacting aminoacids. The authors should provide statistics to validate the structure models they chose and their reliability and also to validate their interaction predictions. When looking at the interaction highlighted, it is mostly driven by loops which are generally intrinsically disordered, therefore hardly predictable. Flawed or too variable initial structure models and subsequent variations in the interaction model itself could generate very different output. Statistics including these parameters and these two levels of variability that can affect the outcome of the analyses must be provided to support the results. In the present stage of the manuscript, although the analyses are sound, they require more support. A comprehensive approach would require taking the top10 models for each protein and make all reciprocal combinations in docking analyses. Only aminoacids appearing in a significant number of interactions should be considered and discussed, with the proper statistics associated.

Another bias to be addressed is the lack of data to demonstrate the link between the binding ability to chitin and the enzymatic inhibition on one side and the higher mortality and decreased weight on the other side. That ApKTI is deleterious is demonstrated but no data help the reader to determine whether it is only its enzymatic activity, only its chitin-binding properties or both (and if both, which participates to which percentage). First, the authors should show that the enzyme inhibition and chitin-binding observed in vitro is also true in vivo. The two following experiments would validate in vivo the in vitro measurements. They could measure the enzymatic activity in larval gut after sublethal exposure to the compound to demonstrate the inhibitory activity in vivo, which would corroborate the in vitro analyses. In addition, they could label the proteins (commercial kits exist to label purified proteins) and perform fluorescent microscopy to see if labelled proteins are localized preferentially on the gut lining (which would suggest that they really have a chitin-binding activity in vivo) or if they are homogeneously distributed within the gut. However, this would not support the causal link. The only way that I see to validate the causality is to perform mutations in the chitin-binding pocket, the enzymatic active site and both the two to abrogate their function and address the role of each in the observed phenotype. This would require producing the enzyme recombinantly though. The latter experiment is time consuming and by doing so, the authors might want to consider another journal. If the authors do not want to perform the last experiment, they must rewrite their manuscript to erase any causal links that are not supported by the results and replace by “suggests” and synonyms.

Below are more specific comments, in chronological order:

Line 53: “such” is missing before “as”

Lines 55-56: sentence is weirdly constructed. The role of larvae is basically to eat to store enough resources for proper pupation and adult emergence. Therefore, blocking/impairing digestive enzymes is a sound strategy that has been investigated for pest control.

Line 57: This statement is not true (and not consistent with the paragraph lines 70-76 on the PM).

Line 76: replace IPs by PIs

Lines 91-98. The description of the fractions is not very clear. The figure isn’t either. Considering that the MM is at the end of the article, a short description of what has been done and a better annotation of the figures is mandatory to make it crystal clear for the reader.

Line 112-113: which essay are you talking about? I did not see the data.

Line 125: replace elucidated by suggested

Figure 3: Has this been replicated? There does not seem to be any error bars. Also, why are the curves not starting at 0? Blank should be subtracted.

Line 190: the “k” is not in capital in ApKTI.

Line 297: space missing between tetraborate and 50

Line 300: The molarity of HCl used for elution is not consistent with the legend of Figure 1. Same comment line 307.

Reviewer 2 Report

Plodia interpunctella or grain moth is a common grain-feeding pest found around the world, consuming cereals or fruits, with larvae capable of biting through plastic and cardboard infesting closed packages. The manuscript by de Oliveira et al. presents the insecticidal effects of a Kunitz trypsin inhibitor from Adenanthera pavonina seeds, against P. interpunctella. It raises the topic of searching for new bioinsecticides, which is quite popular nowadays, and learning about their effects on insect physiology. The authors present a wide range of results, from kinetic studies showing that ApKTI is a non-competitive inhibitor for trypsin and chymotrypsin, bioinformatic modeling and the mortality tests. In my opinion, this is globally a nice and original piece-of-work. It appears to be scientifically sound, however some points need to be corrected before publication. The manuscript seems to be well-written and clear, however I am not a Native English speaker and I do not feel qualified to judge the language.

The authors say in M&M section that the larvae were weighed for determination of WD50 and LD50. However nowhere in the manuscript can be found this information. One has to guess the values from the Figure 5. When calculating the LD50 values the authors should fit no-linear regression curve (sigmoidal) instead of linear (Fig. 5). Even the connecting lines between the points on that graph “try to” align them selves to sigmoidal curve. That way the calculations of LD50 will be done correctly.

I am not convinced to the number of replicates done in mortality tests. Overall the number of insects in one dose is n=9. I suggest to extend the number of replicates from 3 to at least 5 (n=15) to determine LDx values more accurately and precisely. This is quite easy and does not require a lot of work.

Fig. 5 – please change “Representative image of larvae of each treatment, in sequential order:” to “Representative image of larvae of each treatment, in sequential order from the left:”

In my opinion the highlights should be rewritten i.e. “ApKTI increases mortality of P. interpunctella larvae”. These should be bullet points that capture the novelty of Your research and provide readers with an “at-a-glance” overview of the main findings of your article.

Minor comments:

Line 41 (if not re-written) - change Plodia interpunctela to Plodia interpunctella Line 138 – change “studied” to “study” Line 297 and following lines – change 40 mL.h-1 to 40 mL/min (the way of writing chosen by the authors seems a bit unnatural to me) Line 347 – change Interpunctella to P. interpunctella

Overall, the manuscript explores interesting topic and I believe that after some modifications it can be published in the journal Molecules

Reviewer 3 Report

The paper contains interesting data, which can attract readers. I suggest publication after minor corrections. 

First of all, the title is not fully informative. The manuscript contains purification of trypsin inhibitor as well as structural studies. In my opinion, they should be included in the title. Otherwise, the title suggests that the research focuses only on insecticidal activity.

Next, capture to Fig. 1 does not describe all the dotted lines. Therefore, the figure is not self-explanatory. 

The results for the subchapter 2.2 does not refer to the figure. 

Figure 5. What do the error bars at A? Why there are bars for 1.5 not present? Why results for B do not possess error bars? Were they so small?

Lone 338: "After this period, the larvae were weighed for determination of WD50 and LD50." - the results are not shown. I suggest placing them at Fig. 5.

Reference 28 does not contain page numbers. It appeared in 2015, not 2014 (according to NCBI).

Reviewer 4 Report

I have read with interest the MS titled Dual Insecticidal Effects of Adenanthera pavonina  Kunitz-Type Inhibitor on Plodia interpunctella is Mediated by Digestive Enzymes Inhibition and  Chitin-Binding Properties, in my opinion the MS is robust and it aproach is correct. I my opinion few modification are needed before its pubblication.

the botanical authority have to be mentioned at the first appearence of the scientific names in the text

the origin, the conservation and the use of A. pavonina  seeds  have to be described in details

Highlights have some errors, correct it

references have to be correctly formatted

Round 2

Reviewer 1 Report

The authors answered convincingly to my comments. Although the authors did not add new experiments, they addressed the points raised and the manuscript is now better organized and more balanced.